# Evidence-Based Severity Assessment of Animal Models for Pancreatic Cancer

**DOI:** 10.3390/biomedicines12071494

**Published:** 2024-07-05

**Authors:** Tim Schreiber, Ingo Koopmann, Jakob Brandstetter, Steven R. Talbot, Lea Goldstein, Lisa Hoffmann, Anna Schildt, Markus Joksch, Bernd Krause, Robert Jaster, Rupert Palme, Dietmar Zechner, Brigitte Vollmar, Simone Kumstel

**Affiliations:** 1Rudolf-Zenker-Institute of Experimental Surgery, Rostock University Medical Center, 18057 Rostock, Germany; tim.schreiber@med.uni-rostock.de (T.S.); ingo.koopmann@uni-rostock.de (I.K.); jakob.brandstetter@uni-rostock.de (J.B.); lea.goldstein@uni-rostock.de (L.G.); lisa.hoffmann@uni-rostock.de (L.H.); dietmar.zechner@uni-rostock.de (D.Z.); brigitte.vollmar@med.uni-rostock.de (B.V.); 2Institute for Laboratory Animal Science, Preclinical Data Science, Hannover Medical School, 30625 Hannover, Germany; talbot.steven@mh-hannover.de; 3Core Facility Multimodal Small Animal Imaging, Rostock University Medical Center, 18057 Rostock, Germany; anna.schildt@med.uni-rostock.de; 4Department of Nuclear Medicine, Rostock University Medical Center, 18057 Rostock, Germany; markus.joksch@med.uni-rostock.de (M.J.); bernd.krause@uni-rostock.de (B.K.); 5Division of Gastroenterology, Department of Medicine II, Rostock University Medical Center, 18057 Rostock, Germany; robert.jaster@med.uni-rostock.de; 6Unit of Experimental Endocrinology, Department of Biological Sciences and Pathobiology, University of Veterinary Medicine, 1210 Vienna, Austria; rupert.palme@vetmeduni.ac.at

**Keywords:** pancreatic cancer, animal models, refinement, severity assessment

## Abstract

Animal models are crucial to preclinical oncological research and drug development. Animal experiments must be performed in accordance with the 3R principles of replacement and reduction, if possible, and refinement where these procedures remain crucial. In addition, European Union legislations demand a continuous refinement approach, as well as pro- and retrospective severity assessment. In this study, an objective databased severity assessment was performed in murine models for pancreatic cancer induced by orthotopic, subcutaneous, or intravenous injection of Panc02 cells. Parameters such as body weight change, distress score, perianal temperature, mouse grimace scale, burrowing, nesting behavior, and the concentration of corticosterone in plasma and its metabolites in feces were monitored during tumor progression. The most important parameters were combined into a score and mapped against a reference data set by the Relative Severity Assessment procedure (RELSA) to obtain the maximum achieved severity for each animal (RELSA_max_). This scoring revealed a significantly higher RELSA_max_ for the orthotopic model than for the subcutaneous and intravenous models. However, compared to animal models such as pancreatitis and bile duct ligation, the pancreatic cancer models are shown to be less severe. Data-based animal welfare assessment proved to be a valuable tool for comparing the severity of differently induced cancer models.

## 1. Introduction

Animal models have contributed enormously to oncological research [1,2,3,4,5,6]. Pancreatic cancer is the fourth leading cause of cancer death, with a 5-year survival rate of 12% [7]. The poor prognosis highlights the need for continuous research on this pathology. Over the past decades, numerous murine models for pancreatic cancer have been established, such as implantation models, spontaneouse tumor models, and genetically engineered models, with these models serving as indispensable tools for understanding disease progression and pathophysiology, as well as for the evaluation of therapeutic approaches [8,9]. When conducting in vivo research, authorities demand appropriate animal welfare, strictly regulated by national and international law. Animal experiments generally have to be planned according to the 3R principles (replace, reduce, refine), and an analysis of potential harm and benefit must be performed in advance [10,11].

Directive 2010/63/EU obliges researchers in the European Union’s member states to perform a pro- and retrospective severity assessment by assigning animal research procedures to non-recovery, mild, moderate, and severe categories. However, preclinical oncological animal models are listed in each category [12]. These circumstances hinder scientists from performing a reasonable prospective severity assessment of their specific cancer model. Classifying the different severity categories by merely subjective estimations is also questionable. Data-based animal welfare assessment is the key to facilitating an objective and realistic severity grading of each animal model.

The welfare of an animal during different procedures and also during cancer progression might be influenced by distress, pain, or anxiety. Therefore, many different parameters are necessary to reflect the overall burden on an animal [13]. Clinical parameters such as body weight change, clinical scores, body temperature, and mouse grimace scale (MGS) proved sensitive enough to detect distress and pain in different murine in vivo models [14,15,16]. Behavioral parameters like burrowing and nesting activity mirror activities of daily living in humans [17]. A reduction in these parameters indicated an impairment of welfare in mice and was already quantified after surgical interventions and during the progression of chronic diseases [17,18,19]. The hormone corticosterone is an important indicator of rodent stress [20]. Analyzing the fecal corticosterone metabolites (FCM) represents a non-invasive method to assess the endocrine stress responses [21].

The aim of the present study was a data-based severity assessment in different cell-derived murine cancer models. Murine pancreatic cancer cells were either injected orthotopically into the pancreas, subcutaneously into the flanks of the mice, or intravenously to induce lung metastases. These injection sites cover the commonly used cell-derived animal models in preclinical oncological research [22,23,24]. The above-mentioned clinical, behavioral, and hormonal parameters longitudinally assessed the welfare of the mice. The quantitative outcome measures of body weight, burrowing, and the clinical score were used to obtain a relative metric for severity comparison with the RELative Severity Assessment (RELSA) procedure [25]. This computational approach directly compares the evidence-based severity between the cancer models. A direct grading of severity with other animal models will also be possible. Data-based welfare assessment and the RELSA procedure can also be used to analyze the effect of refinement methods such as analgesia. Further, the above-mentioned parameters revealed that high doses of tramadol negatively affect animal wellbeing compared to low doses of tramadol or buprenorphine in a murine osteotomy model [26]. Definitions of early humane endpoints can be articulated in a data-based manner with the parameters used, as already described in a murine bile duct ligation model, where a reduction in burrowing behavior and body weight reduction of >10% could predict the endpoint of the mice within two days [27]. These studies indicate that a data-based severity assessment might play an important role in the implementation of refinement methods in future animal studies and improve the definition of suitable early humane endpoints.

## 2. Materials and Methods

### 2.1. Cells

The National Cancer Institute provided the murine Panc02 cells. The cells were cultured in RPMI-1640 medium (Sigma-Aldrich, St. Louis, MO, USA, supplemented with 10% fetal calf serum, penicillin, and streptomycin).

### 2.2. Animals

The German local authority Landesamt für Landwirtschaft, Lebensmittelsicherheit, und Fischerei Mecklenburg-Vorpommern (7221.3-1-010/21) approved all animal experiments in accordance with the German animal protection law and the European Directive 2010/63/EU [12]. Breeding pairs of C57BL/6J mice were purchased from Charles River Laboratories and bred in our facility at the Rostock University Medical Center under specific pathogen-free conditions. During the experiment, the mice were kept single-housed in type III cages (Zoonlab GmbH, Castrop-Rauxel, Germany) under a 12 h dark/light cycle, a temperature of 21 ± 2 °C, and relative humidity of 60 ± 20%, with food (pellets, 10 mm, ssniff-Spezialdiäten GmbH, Soest, Germany) and tap water ad libitum. Enrichment was provided in the form of nesting material (shredded tissue paper, Verbandmittel GmbH, Frankenberg, Germany), paper rolls (75 × 38 mm, H 0528–151, ssniff-Spezialdiäten GmbH), and wooden sticks (40 × 16 × 10 mm, Abedd, Vienna, Austria).

### 2.3. Animal Models for Pancreatic Cancer

For the cell line-based PDA models, nine mice of both genders were used for the orthotopic, intravenous, or subcutaneous injection. The number of animals was previously approved by the local authorities for these experiments (7221.3-1-10/21).

Before each tumor cell injection, all animals were anesthetized with 1–3 vol. % isoflurane. The mice were placed on a heating plate at 37 °C, and eyes were kept wet by eye ointment.

Four male and five female C57BL/6J mice (age: 20–21 weeks) with an average body weight of 24.9 g (21.1–28.0 g) were used for the orthotopic tumor cell injection. A single subcutaneous injection of 5 mg/kg carprofen (Rimadyl, Pfizer GmbH, Berlin, Germany) was applied as analgesia before surgery. The abdomen of the mice was shaven and disinfected before opening the abdominal cavity by laparotomy. A total of 5 µL of the cell suspension (1 × 10^4^ cells of the murine cell line Panc02 in 1:1 PBS/matrigel) was slowly injected with a 25 µL syringe (Hamilton, Reno, NE, USA) into the pancreas head. The pancreas was put back into the cavity, and the peritoneum was closed by a continuous surgical suture with a coated 5-0 Vicryl suture (Johnson & Johnson Medical GmbH, New Brunswick, NJ, USA). The subcutaneous skin was sewn separately with 3–5 surgical knots by using a 5-0 Prolene suture (Johnson & Johnson Medical GmbH, New Brunswick, NJ, USA).

Four male and five female C57BL/6J mice (age: 16–20 weeks) with an average body weight of 25.5 g (20.6–29.3 g) received subcutaneous injections of the tumor cells. After placing the mice ventrally on the heating plate, the fur at the flanks of the mice was shaven. For each mouse, 5 × 10^5^ Panc02 cells in 100 µL PBS were injected into the left and right flank with a 300 µL syringe (Becton, Dickinson and Company, Franklin Lakes, NJ, USA), respectively. The tumor growth was measured two times a week throughout the experiment via caliper. For this procedure, the mice were kept under short isoflurane anesthesia (1–2 vol. %, 2–3 min) on a heating plate at 37 °C. Assuming the hemi-ellipsoidal shape of the tumors, the volume (V) was calculated by length (L) and width (W) using the following formula [28]:V = 0.52 × L × W^2^

For the intravenous injection of the tumor cells, five male and four female C57BL/6J mice (age: 18–21 weeks) with an average body weight of 24.3 g (18.1–33.0 g) were used for the intravenous injection. One female mouse did not show any tumor cell engraftment in the lung, and the data of this mouse were therefore excluded from the study. The mice were anesthetized as described above. For injection, mice were positioned on their side on a heating plate at 37 °C, and 1 × 10^6^ cells in 50 µL PBS were injected into the lateral tail vein using a 30 G needle (BD, Franklin Lakes, NJ, USA) and a catheter (ICU Medical, Inc., San Clemente, CA, USA).

Regardless of the type of injection, all mice were placed in front of a heating lamp for recovery after tumor cell injection. Continuous analgesia was provided for all mice by daily application of metamizole (3 mg/mL, Novaminsulfon-ratiopharm^®^, Ulm, Germany) in the drinking water. The analgesic treatment started one week before each tumor cell injection until the end of the experiment.

### 2.4. Animal Models for Gastrointestinal Diseases and Transmitter Implantation

To compare the severity of the cancer models with other animal models for gastrointestinal diseases, the variables (body weight, burrowing behavior, and distress score) were obtained from our previous studies to calculate the RELSA_max_ [29,30,31,32,33,34]. The induction of liver damage was performed on 22 male BALB/cANCrl mice. To induce the liver damage, carbon tetrachloride (Merck Millipore, Eschborn, Germany) was diluted fourfold with corn oil (C8267, Sigma-Aldrich, St. Louis, MO, USA) and i.p. injected at a specified dosage (0.25 mL/kg) twice per week over 6 weeks. The NLRP3-Inhibitor MCC950 (20 mg/kg; Sigma Aldrich, n = 13) or aqua dest. (n = 9) was i.p. injected daily from day 28 to 41. The methodological details are published elsewhere [30]. As described above, the orthotopic pancreatic cancer model was established on 26 male C57Bl6/J mice. Instead of the Panc02 cells, 2.5 × 10^5^ murine 6606PDA cells were injected into the pancreas. One subgroup of mice had an ETF-10 Transmitter implanted 14 days before tumor cell injection (n = 7) [32]. Two subgroups of mice were treated with therapeutic interventions from day 4 until day 37 of tumor progression either by daily i.p. injections of α-cyano-4-hydroxycinnamate (CHC, 15 mg/kg; Tocris Bioscience, Bristol, UK, n = 7) in combination with metformin (daily, i.p, 125 mg/kg; Merck, Darmstadt, Germany), galloflavin (GAL; three times a week, i.p., 20 mg/kg, Tocris Bioscience, n = 7) plus metformin, or with corresponding vehicles (vehicle (MET + CHC) n = 7; vehicle (MET + GAL) n = 5) [31]. Chronic pancreatitis was induced on 16 male C57BL6/J mice by repetitive cerulein injections (50 µg/g, i.p., 3 hourly injections/day on 3 days a week, for 4 weeks, Bachem, H-3220.0005, Bubendorf, Switzerland). As a therapeutic intervention, MicroRNA21-inhibitor (miRCURY LNA^TM^microRNA microRNA-21a-5p inhibitor; sequence: TCA GTC TGA TAA GCT) and its control (miRCURY LNA^TM^microRNA microRNA-21a-5p inhibitor control; sequence: TCA GTA TTA GCA GCT), 10 mg/kg, s.c.; Qiagen, Hilden, Germany) were injected on day 0 until day 14 after the first cerulein injection, inhibitor (n = 8) vehicle (n = 8) [33]. For the induction of bile duct ligation, a laparotomy was performed on 30 male BALB/cANCrl mice under isoflurane anesthesia and carprofen analgesia (5 mg/kg, i.p.). The bile duct was ligated by three surgical knots and transected between the two distal ligations. Therapy was conducted with MCC950 (n = 14) or the corresponding vehicle (n = 16) from day 1 before BDL until day 13 after BDL [30]. The data of mice after transmitter implantation were used as the reference set. Transmitters were implanted into the abdominal cavity of 10 male C57Bl6/J mice under isoflurane anesthesia and carprofen (5 mg/kg, i.p.) analgesia, as described previously [32]. For all animal models, continuous analgesia was provided by metamizole (1250 mg/L; Ratiopharm, Ulm, Germany) in the drinking water.

### 2.5. PET-CT Imaging

To quantify tumor progression in vivo, animals were anesthetized with 1.0–2.5 vol.% isoflurane, and eye ointment was applied to prevent dehydration. PET-CT imaging with the radiotracer 2-[^18^F]fluoro-2-deoxy-D-glucose ([^18^F]FDG) was performed in mice undergoing intravenous or orthotopic cell inoculation. The mice were injected intravenously with ~15 MBq of [^18^F]FDG in the tail vein under isoflurane anesthesia. At 1 h after injection, static PET scans in the head-prone position were recorded for 15 min (Inveon PET-CT Siemens, Knoxville, TN, USA). Throughout the entire imaging procedure, the body temperature was kept stable using a heating pad. The PET-image reconstruction method comprised a two-dimensional ordered subset expectation maximization (2D-OSEM) algorithm with four iterations and six subsets. Attenuation correction was achieved with the whole-body CT scan, and a decay correction was applied. PET images were corrected for random coincidences, dead time, and scatter. The [^18^F]FDG uptake (% injected dose/g body weight) in each tumor was quantified as metabolic tumor volume (30% of the hottest voxel) [26] using the program Inveon Research Workplace (Siemens Healthcare AG, Zurich, Switzerland).

### 2.6. Histology

To evaluate the number and size of metastasis, lung tissue was fixed with 4% paraformaldehyde, sliced in 4 μm sections, and dried at 65 °C for 2 h. Afterwards, the slices were stained with hematoxylin and eosin (H&E). The number of metastases and the metastatic area/lung slice were quantified with the program QuPath 0.4.3 [35].

### 2.7. Assessment of Clinical Parameters

To analyze animal distress upon laparotomy and tumor cell injection for all pancreatic cancer models, all distress parameters were assessed before (pre) and daily directly after operation (op) until recovery day 3. To analyze the course of distress, all parameters were additionally quantified during the early, middle, and late phases of tumor progression, respectively, for each cancer model and on the final days before reaching the humane endpoint (hEP, Figure 1). The distress analysis before humane endpoint evaluation started for each animal, when a clinical score was quantified according to the score sheet (Appendix A), or when a body weight loss of more than 5% compared to the previous day was observed. This schedule proved helpful in earlier projects, enabling the comparison of distress between different animal models [29]. The body weight was measured daily, and body weight change [%] was calculated referring to one day before tumor cell injection in the healthy mice. The distress score was assessed daily by observation of the mice for 1–3 min in their home cages, according to our clinical score sheet (Appendix A). The surface temperature of mice was measured perianally via non-contact infrared thermometry (WEPA Apothekenbedarf GmbH & Co. KG, Hillscheid, Germany) as previously described by Mei et al. [15].

For the evaluation of pain behavior, the assessment of the Mouse Grimace Scale (MGS) was performed, according to Langford et al. [36]. Mice were transferred into a colorless transparent polycarbonate box (9 × 5 × 5 cm). The box was placed inside an illuminated white light tent and additionally lighted from the front. Mice were allowed to acclimate in the box for 5 min before they were filmed for 5 min with a digital single-lens reflex camera (Canon EOS 70D, Tokyo, Japan). Afterwards, 3 pictures were screenshotted for each mouse per time point, randomized, and scored by three blinded researchers as described previously [36]. The MGS score was calculated by averaging the three independent scorings and normalizing the MGS score for each mouse to the pre-operative phase.

### 2.8. Assessment of Behavioral Parameters

The burrowing was analyzed according to Deacon [17]. A burrowing tube (15 × 0.03 × 6.5 cm) was filled with 200 g food pellets (ssniff-Spezialdiäten GmbH, Soest, Germany) and placed in the cage 2.5–3 h before the dark phase. After 2 h, the weight of pellets displaced from the tube was calculated, and additionally, on the following day, the weight was calculated after 17 h. Animals that burrowed less than 100 g pellets within 2 h and less than 150 g overnight during the pre-phase were excluded from the final data analysis. According to the above-mentioned thresholds, the following animals were excluded from the 2 h burrowing: orthotopic: n = 3; subcutaneous: n = 2; intravenous: n = 2. For the 17 h burrowing, one mouse was excluded for the orthotopic and subcutaneous models, respectively.

To evaluate the nesting behavior, a nestlet (5 cm square of pressed cotton batting, ZOOONLAB GmbH, Castrop-Rauxel, Germany) was supplied for each mouse 1–2 h before the dark phase, and on the following day, the score was assessed according to the 1–5 point scale from Deacon [37]. In addition to the score from Deacon, a score of 6 was applied to a perfect nest (90% is torn) resembling a crater, where more than 90% of the circumference of the nest wall is higher than the body height of the mouse. The final data analysis excluded animals that did not reach a nesting score of 4 during the pre-phase. According to this threshold, the following animals were excluded for the parameter nesting: orthotopic: n = 4; subcutaneous: n = 2; and intravenous: n = 1. Since it is reported that mice can learn from each other [17], both behavior assays were performed two times in group housing during the early pre-phase, one week before each tumor cell injection for the PDA models and one week before induction of the gastrointestinal diseases or transmitter implantation. Afterwards, all mice were housed individually until euthanasia. All the mice for the cell-line-based PDA models were euthanized when humane endpoint criteria were noticed according to the score sheet (Appendix A). Mice from the 6606PDA model, the transmitter implantation, or the other gastrointestinal animal models were euthanized at the end of each experiment. Animals that reached the humane endpoint during the experiment according to the score sheet criteria were excluded from the analysis of RELSAmax for the gastrointestinal animal models. No mouse was found deceased during the study, since the health of mice was observed on a daily basis to identify euthanasia criteria early.

### 2.9. Analysis of Hormonal Parameters

Corticosterone concentrations were measured in blood plasma and its metabolites in feces (FCMs). Feces were collected from home cages 24 h after cage change, dried for 4 h at 65 °C, and stored at −20 °C. Further, 50 mg of the dry feces was extracted with 1 mL 80% methanol for subsequent analysis using a 5α-pregnane-3β,11β,21-triol-20-one enzyme immunoassay [38].

Blood was collected by a retro-orbital puncture after 2–3 min anesthesia with isoflurane (4%). A fast sampling procedure is required to avoid any influence of the sampling methods on the corticosterone level. The blood samples were centrifuged (1200× *g* for 10 min at 4 °C), and blood plasma was stored at −80 °C. According to the manufacturer’s instructions, plasma corticosterone concentrations were measured using an ELISA Kit (DEV9922, Demeditec Diagnostics GmbH, Erfurt, Germany).

### 2.10. Data Analysis

All data were graphed and analyzed using GraphPad Prism 8.4.3 (GraphPad Software, San Diego, CA, USA). For Figure 2, Figure 3 and Figure 4, the data are presented as box plots indicating upper and lower quartiles and the min–max range of all data points. A repeated measures two-way ANOVA or mixed-effects model with Geisser–Greenhouse correction for sphericity control was applied, followed by Tukey’s test to compare the three models. Dunnett’s multiple comparisons test was performed, allowing testing for significant differences detectable by longitudinal assessment within the specific models compared to pre-interventional data. Differences with *p* ≤ 0.05 were considered significant. Statistical details of the ANOVAs performed are listed in Appendix A.

The severity of the animal models (Figure 5) and the gender comparisons within each model (Appendix A) were calculated with the Relative Severity Assessment (RELSA) algorithm [25] from the RELSA R-package (https://talbotsr.com/RELSA/index.html, accessed on 15 February 2024). The RELSA score was determined with three input variables (body weight, burrowing activity, distress score) from the following time points for each animal model: pre, op, early, middle, and late phases of tumor progression. These variables were mapped against reference data from transmitter implantation in mice with the same variables from previous studies [29,32]. The highest RELSA value (RELSA_max_) represents the highest distress during tumor cell injection and progression. The determined RELSA_max_ values for each animal model were 10,000-fold bootstrapped to obtain estimates on the median (Ȓ) and the 95% confidence intervals. The Kruskal–Wallis test calculated significant differences in the RELSA_max_ values between the animal models. Sex-related differences in the RELSA_max_ values within each model were analyzed by ANOVA (Appendix A).

## 3. Results

All tumor models were analyzed for sex-specific survival and cancer burden differences (Appendix A). No significant differences were found for survival and model-specific tumor burden criteria except for the final tumor weight in the orthotopic model (Appendix A). Nevertheless, the 30% metabolic tumor volume, quantified by the uptake of [^18^F]FDG of either sex, showed no difference in the orthotopic model, measured at the same time of tumor progression (Appendix A). Sex-specific differences were also quantified for the animal welfare parameters during the operative phase (Appendix A). At a few time points (7 out of 120 over all models and parameters), significant sex-specific differences were detected for parameters such as perianal temperature, burrowing behavior, and FCM (Appendix A). However, since mostly no sex differences were detected, we pooled the data for male and female mice for the severity assessment analysis.

### 3.1. Severity Assessment Post Tumor Cell Injection

The assessment of all non-invasive animal welfare parameters was performed before any intervention (pre), on the day of cell inoculation (0), and on the following 3 recovery days (1–3) to detect distress caused by the intervention (Figure 2 and Appendix A). On the day of tumor cell inoculation, a significant reduction in body weight compared to the pre-value and to the other two models was detected for the orthotopic model (Figure 2A). No significant changes were detected after tumor cell injection (Figure 2B) for the distress score. After subcutaneous cell injection, mice displayed a significantly higher perianal temperature on the first recovery day than those intravenously inoculated (Figure 2C). For animals of the orthotopic and intravenous model, a temporary, non-significant decrease in the nesting activity was observed after cell injection (Figure 2D). The burrowing activity within 2 h was significantly decreased on the day of surgery after orthotopic injection and on recovery day 2 and 3 post intravenous cell application (Figure 2E). However, no significant changes in burrowing behavior were observed after 17 h, except for the intravenous model on day 3 post injection (Figure 2F). No significant changes were detected after cell injection in either model for the mouse grimace scale.

Nevertheless, a non-significant temporary increase in the MGS score on the day of cell implantation was seen for mice of the orthotopic model (Figure 2G). The concentration of FCM was significantly increased on the day of surgery until recovery day 1 in the orthotopic model. A significant increase in FCM concentration was also observed after subcutaneous injection on recovery day 1 (Figure 2H).

### 3.2. Animal Welfare Assessment during Tumor Progression

Parameters for severity assessment were monitored to detect tumor progression-related changes in distress shown for early, middle, and late phases, respectively, for each model (Figure 3). During tumor progression, the body weight of the mice increased over time for all animal models (Figure 3A). A significant increase in body weight of the intravenous model was detected in the middle phase when compared to baseline and to the orthotopic model (Figure 3A). For the subcutaneous model, a significant body weight change was observed in the late phase (Figure 3A). No significant changes were noticed for the distress score nor for perianal temperature during tumor progression (Figure 3B,C). In all three models, a discreet decrease in nesting activity during tumor progression was observed. However, a significant reduction was only seen in the late phase for the subcutaneous model (Figure 3D). Mice of the orthotopic and intravenous model showed similar burrowing behavior after 2 h compared to the baseline values of healthy mice (Figure 3E). Moreover, the burrowing behavior of the subcutaneously injected animals was significantly increased in the middle and late phase of tumor progression (Figure 3E). No significant differences were observed for burrowing behavior after 17 h for each model (Figure 3F). The concentration of corticosterone in plasma and its metabolites in feces did not change significantly due to tumor progression compared to the baseline (Figure 3G,H). However, it has to be mentioned that a significant difference in fecal corticosterone metabolite concentration was observed between the orthotopic and the intravenous model for baseline values (Figure 3G). No significant changes in the MGS scores were observed for the tested models during tumor progression (Figure 3I).

### 3.3. Severity Assessment for Humane Endpoint Determination

Starting from the middle phase of tumor progression, animals with an increased distress score or a body weight loss of more than 5% from the previous day underwent a daily welfare assessment until criteria for timely euthanasia occurred according to the score sheet (Appendix A). No significant changes in body weight were noticed for the subcutaneous and intravenous models until the day of the humane endpoint compared to baseline values (Figure 4A). However, body weight decreased within the 4 days until euthanasia in the orthotopic model, resulting in a significant body weight reduction on the day before and the day of the humane endpoint. The body weight loss of the orthotopic mice resulted in significant differences in both models three days before the humane endpoint and a significant difference on the day of euthanasia from the subcutaneously injected mice (Figure 4A). For all models, the distress score increased within the last two days until the individual endpoint was noticed (Figure 4B). On the day of euthanasia, subcutaneously injected animals had a significantly lower distress score when compared to the orthotopic and intravenous models (Figure 4B). Animals of the intravenous model had significantly decreased perianal temperature at the individual humane endpoint (Figure 4C). The behavioral parameters of nesting and burrowing were reduced on the day of euthanasia and the day before in all models (Figure 4D–F). Moreover, the amount of burrowed pellets after 17 h in the orthotopically injected mice was significantly reduced and significantly lower on the day of the humane endpoint compared to subcutaneous tumor-bearing mice (Figure 4F). A slight but non-significant increase was noticed for the MGS score until the individual humane endpoints for analyzed animal models (Figure 4G). Additionally, on the day of the humane endpoint, mice with orthotopic and intravenous injected tumor cells had significantly increased concentrations of the stress hormone corticosterone compared to the baseline values. These corticosterone concentrations were significantly higher than the measured concentrations in subcutaneously injected mice (Figure 4H).

### 3.4. Between-Model Comparison of Severity by RELSA_max_ Analysis

Multiple variables were combined into RELSA scores to enable a data-based comparison of severity between the cancer models [25]. Therefore, the sensitive severity parameters of body weight change, burrowing behavior for 2 h, and distress score were used from the day before any intervention (pre), directly after tumor cell injection (0), and during early, middle, and late phases of tumor progression. The resulting RELSA scores were mapped against reference data (transmitter implantation in mice) to provide context for quantitative severity. Further, the highest RELSA score was obtained for each animal during tumor cell injection or cancer progression. The animal models were ranked according to the order of their RELSA estimates. The intravenously injected mice (Ȓ = 0.075, CI95% = [−0.049;0.199]) displayed the lowest RELSA_max_ values, followed by the subcutaneous tumor model (Ȓ = 0.177, CI95% = [0.060;0.294]; (Figure 5A)). According to the RELSA_max,_ the highest severity was quantified for orthotopic-injected mice (Ȓ = 0.440, CI95% = [0.323;0.557]). The RELSA_max_ estimates of the orthotopic model were significantly higher than for the intravenous and subcutaneous injected animals (Figure 5A). Testing for sex-specific differences within each cancer model did not show significant differences (Appendix A).

Furthermore, the RELSA_max_ values of the three cancer models were directly compared to the severity of other murine gastrointestinal animal models, such as liver fibrosis induction by CCl_4_, pancreatitis, another pancreatic cancer model, and bile duct ligation. The data for these models were assessed in a former study [29]. The lowest severity was observed for the intravenous and subcutaneous cancer models (Figure 5B). The intravenous model had significantly lower RELSA_max_ values than the liver fibrosis models, orthotopic pancreatic cancer, pancreatitis, and bile duct ligation. Furthermore, the subcutaneous model had significantly lower RELSA_max_ estimates than the orthotopic PDA (6606PDA), pancreatitis, and bile duct ligation models. According to the RELSA algorithm, the orthotopic model with the Panc02 cells was ranked fourth among the seven animal models. The pancreatitis model and the bile duct-ligated mice displayed a significantly higher severity than the orthotopic PDA model in the present study (Figure 5B).

## 4. Discussion

### 4.1. Severity Assessment of Cell-Induced PDA Models

Since mice are prey animals and suppress signs of pain, distress needs to be assessed by several clinical, behavioral, and hormonal parameters to facilitate data-based comparison of severity in different animal models [39,40,41]. The grimace scale is one of the most important tools to assess post-surgical pain in rodents [42]. Body temperature is known to be the most predictive humane endpoint criterion [43,44,45]. The other severity parameters used such as body weight, clinical score, burrowing behavior, nesting activity, and FCM proved to be sensitive enough for severity assessment in animal models of repeated seizures, partial hepatectomy, surgical interventions, and different gastrointestinal diseases [25,29,46,47]. The broad applicability of these parameters indicates that different signs of distress, such as anxiety, depression, pain, harm, or malaise, are recognizable in mice. Using one single parameter for severity assessment might be insufficient to depict the variety of distress [13], especially for comparing different animal models.

Though all PDA models in this study were induced by injection of Panc02 cells, the orthotopic model caused significantly more distress than the subcutaneous and the intravenous one (Figure 5). Until reaching the individual humane endpoint, the distress is mainly caused by the cell injection procedures.

After inoculation of tumor cells in the PDA models, a significant reduction in body weight and burrowing behavior and an increase in FCMs particularly indicated distress in mice of the orthotopic model, whereas these parameters did not change significantly after subcutaneous and intravenous cell injection. This can be explained by a shorter duration of anesthesia for subcutaneous and intravenous cell injection. It was previously reported that inhalation anesthesia using isoflurane with a duration of 45 min, without additional intervention, reduces burrowing behavior and increases MGS score and FCM concentrations [48]. These data are consistent with the findings for the orthotopic model with an anesthesia duration of about 20–35 min. In contrast, only minor changes in these parameters were observed after shorter anesthesia durations for intravenous (5–15 min) and subcutaneous (5–10 min) cell injection. In addition, orthotopic cell injection requires laparotomy. Jirkof et al. showed a reduction in burrowing and nesting behavior for mice that underwent laparotomy compared to control mice undergoing anesthesia only [19,49]. The significant decrease in body weight and burrowing behavior and increased plasma corticosterone concentrations for the orthotopic cell injection might be due to the longer anesthesia duration and the laparotomy.

During the phase of tumor progression with continuous analgesic application via drinking water, none of the severity assessment parameters used indicated increased distress in any of the PDA models (Figure 3). These findings are comparable to the mostly asymptomatic tumor development and progression seen in early-stage cancer patients, which leads to the first diagnosis in the advanced stage (80%) with non-resectable tumors in 80–90% of these cases [50,51,52].

In contrast, the phase of individual humane endpoints is characterized by an impairment of welfare in all animal models. Again, the highest distress was observed in the orthotopic model, indicated by significantly reduced body weight and burrowing behavior, as well as increased distress score and plasma corticosterone concentrations, followed by the intravenous and subcutaneous model (Figure 4). Also, the prolonged survival period after the subcutaneous (~70 days) and intravenous tumor cell injection (~62 days) compared to the orthotopic model (~47 days) indicate a faster tumor progression under orthotopic conditions for the Panc02 cells (Appendix A). Despite the long survival times, we recommend a daily health check of the mice for all kinds of murine cancer models. In the orthotopic model, a significant reduction in body weight and burrowing behavior was noticed even one day before reaching the humane endpoint. Both criteria indicated an early humane endpoint in a mouse model of cholestasis [27]. Body weight alone could predict the individual endpoint for 97% of rats with intracranial glioma [53]. Corresponding with the above-mentioned literature, body weight loss and burrowing behavior might be useful humane endpoint criteria for the orthotopic model. In contrast, for the subcutaneous and intravenous models, no significant changes in these parameters were induced on the days before reaching the humane endpoint. However, burrowing and nesting behavior were reduced in these models, and non-significance might account for to the low n-number, but also for the different observed endpoint criteria for each model according to the score sheet. In the orthotopic model, the mice were euthanized when criteria such as a body weight loss exceeded 15% or a bloated belly and bent body posture were recognized (Appendix A). At the same time, the euthanasia criteria for the subcutaneous model mostly included tumor ulceration. Besides small wounds on the subcutaneous tumors, the mice were still active, and hardly any other criteria from the score sheet were observed. Mice of the intravenous model indicated mostly abnormal breathing behavior as a criterion for euthanasia. These mice also indicated hardly any changes in the general condition or spontaneous or flight behavior (Appendix A). The spontaneous appearance of the above-mentioned model-specific humane endpoint criteria requires a daily health check of the mice for all kinds of murine cancer models. These findings implicate that choosing correct and animal model-specific early humane endpoint criteria substantially reduces long-lasting distress of survival studies in mice. In the future, refinement of the analgesic regime might be useful, since an increased MGS score might indicate pain, especially in the orthotopic model directly after the laparotomy and on the days of the humane endpoint. The use of opioids such as tramadol or buprenorphine proved to be useful as a post-operative analgesia in mice [26]. Another study revealed that buprenorphine had no major impact on tumor growth but promoted the welfare of the animals [54]. Furthermore, cancer patients are usually treated with opioids with increased analgesic potential [55,56]. The use of opioids is, therefore, closer to the clinical setting and might promote animal welfare [49]. Comparable studies using different analgesics and doses are necessary to prove this hypothesis. In addition, in an animal model for DSS-induced colitis, these parameters indicated that paracetamol might be a more favorable analgesic compared to tramadol and metamizole [57].

For the investigated PDA models, no significant sex-specific differences in survival and tumor burden were seen (Appendix A). However, since it is reported that sex can impact engraftment and cause differences in tumor progression [58,59,60], it remains necessary to avoid biased conclusions by analyzing sex specificities in tumor development for each cancer cell line beforehand.

Since the analyzed PDA models show different levels of distress, especially during the phase of humane endpoint, the subcutaneous model could be the one of choice due to the comparatively simple setup and the least overall distress, if the research hypothesis allows it. Subcutaneous models can be used to compare in vitro and in vivo effects in drug development before moving to more complex models [23]. However, subcutaneous tumor models do not allow observations of tumor progression under orthotopic tissue conditions [23]. Further, it was seen that heterotopic models show slower tumor growth, differing involvement of the immune response, and formation of a specific tumor microenvironment compared to orthotopic models, which can even lead to contradicting therapeutical effects in preclinical research, as seen, e.g., in a cell-induced model of lung cancer [61]. Cai et al. showed that depending on the injection site, tumors of a breast cancer cell line developed different profiles of necrosis, inflammation, and vascularization [62]. Reflecting these criteria, the subcutaneous PDA model might not mirror the exact growing environment of tumor origin but can be a first step to transfer in vitro drug testing to an in vivo setting, with less distress for the animals, before investing in more complex models with the occurrence of greater impairment of wellbeing, as seen in the intravenous and orthotopic model. The intravenous PDA model should simulate the development of metastasis. In a murine model for breast cancer metastasis in the lung, it was shown that injection via the tail vein induced tumor cell colonization of the whole lung but no formation of isolated tumors, as seen after orthotopic injection [63]. However, if metastasis follows primary tumor growth, the determining factor will be the primary tumor [64]. Due to ongoing implementation and improvement of surgery in tumor therapy, the intravenous PDA model might simulate a setting to test the potential of new drug candidates for eradication of metastasis after primary tumor resection without determination by growth of the primary tumor. More important than the severity of a specific cancer model should be the correct choice of the animal model to answer the specific research question.

### 4.2. Applicability of the RELSAmax for Severity Assessment

Further, our previous study found varying discriminative power of the measured parameters for different animal models of gastrointestinal diseases [34]. This aspect is also indicated in the present study, since a significant body weight reduction on the day of the humane endpoint can be recognized for the orthotopic model. In contrast, perianal temperature only changed significantly in the intravenous model (Figure 4). Therefore, the complexity of distress in mice requires more than one distress parameter to enable data-based comparison of different animal models [13,34].

Hence, combining widely applicable and sensitive parameters such as burrowing behavior, distress score, and body weight in the calculation of RELSA_max_ enables data-based comparison and delicate ranking of different animal models. Quantifying the highest experienced distress of mice in an animal model, even if it lasts only for a short period of time, RELSA_max_ is consistent with the EU recommendation for retrospective severity assessment [65]. Further, the RELSA has already been used to compare the severity of animal models such as colitis and sepsis [25] and in a spared nerve injury model of neuropathic pain [66].

According to RELSA_max,_ the distress of the PDA models used after injection of tumor cells and during tumor progression was ranked 1, 2, and 4 compared to other animal models of gastrointestinal diseases in an order with increasing severity (Figure 5). In particular, changes in body weight and distress score lead to a significantly higher RELSA_max_ in the animal models for pancreatitis and BDL. For the tested PDA models, the highest average decrease in body weight before the individual endpoint phase was seen in the orthotopic model (−3.6 ± 2.8%) on the day of cell implantation, whereas the intravenous and the subcutaneous model caused an increase rather than decrease in body weight. In contrast, mice suffering from chronic pancreatitis had a maximum average body weight decrease of 5.6 ± 3.3%, and those undergoing BDL of 11.4 ± 6.0% [29]. The body weight reduction, in combination with a higher distress score (6.6 ± 2.4), leads to the significantly higher RELSA_max_ values for the BDL model.

The comparison of the two cell-induced orthotopic PDA models revealed similar RELSA_max_ values. Data of the slightly higher-ranked PDA model induced by 6606PDA cell injection includes transmitter-bearing mice and different therapeutic treatments [29]. The somewhat higher RELSA_max_ may indicate additional distress caused by the variations [29]. However, adding these variables did not cause a significant difference in RELSA_max_ for these two PDA models, thereby confirming the previously described robustness of RELSA_max_ against variations of the experimental setting within one animal model [29].

The implantation of a transmitter was used for reference data for the RELSA_max_ and is set to a value of 1.0. According to the EU guidelines, this surgical procedure has to be classified as moderate, suggesting that the severity of the PDA models with lower RELSA_max_ values can be assorted to the severity grading of mild to moderate until the late phase of tumor progression (intravenous: 0.075, subcutaneous: 0.177, orthotopic: 0.440). This classification of the intravenous PDA model contradicts, in a data-based manner, the EU recommendation for assuming that metastasizing tumor models need to be classified as severe [12]. This suggests that retrospective data-based severity assessment used for the realistic classification of specific animal models could contribute to an evidence-based approval process for animal experiments. A limitation of the current study is that the severity comparison of the PDA models and other gastrointestinal models with the RELSA_max_ was performed in the tumor progression phase only, excluding the humane endpoint phase. The animals for the gastrointestinal diseases were euthanized at the end of each experimental protocol, without any signs of severe suffering. To exclude this bias, the data applied for the RELSA_max_ were taken from the tumor progression phase only. The expected severity of the PDA models at the humane endpoint phase should be moderate to severe. Another limitation of the RELSA_max_ is that this procedure is able to quantify the maximum experienced distress of an animal at a specific time point. However, a possible accumulation of distress over a distinct period cannot be quantified with the RELSA_max_.

## 5. Conclusions

In summary, a databased animal welfare assessment followed by a mathematical scoring with the RELSA_max_ proved to be a suitable tool to describe the severity of differently induced pancreatic cancer models. The orthotopic cancer model indicated the highest distress compared to the subcutaneous and intravenous models. However, the PDA models proved to be less stressful than other gastrointestinal models until the late phases of tumor progression. Despite the varying severity of differently induced cancer models, the correct choice of an animal model should rely on the specific research question and the optimal transfer from bench to bedside.

## Figures and Tables

**Figure 1 biomedicines-12-01494-f001:**
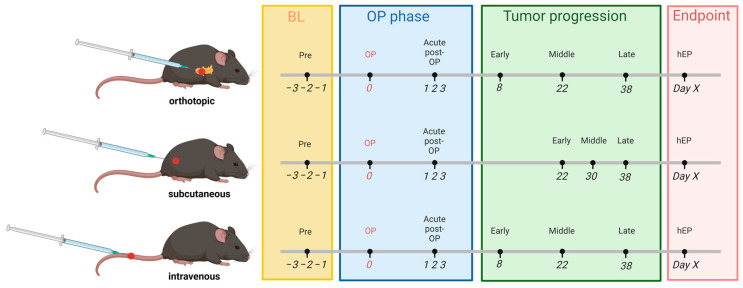
Experimental design for longitudinal evidence-based animal welfare assessment in cell line-derived cancer models. Distress assessment was performed on days −3 to −1 before any intervention (Pre) to determine the baseline values (BL) of healthy mice at the day of cancer cell injection (day 0, OP), on post-operative days 1–3, and during the early, middle, and late phase of tumor progression and on the days before the individual humane endpoint. Created with BioRender.com.

**Figure 2 biomedicines-12-01494-f002:**
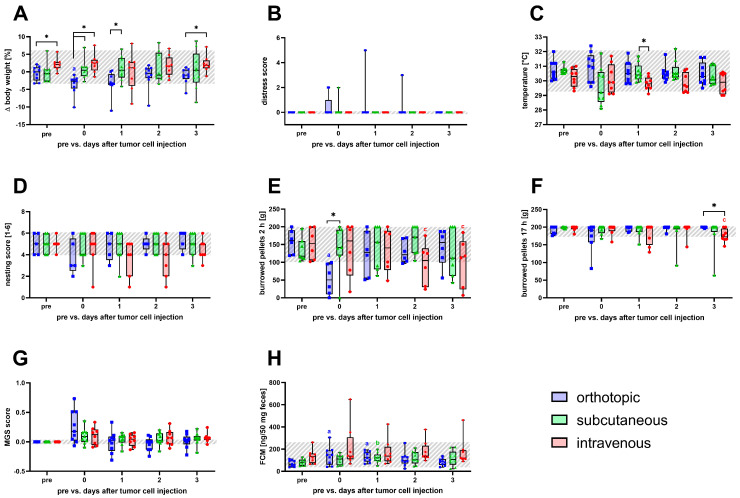
Animal distress was quantified before and after the orthotopic, subcutaneous, and intravenous cell inoculation. Body weight change (**A**), distress score (**B**), perianal temperature (**C**), nesting (**D**), burrowing after 2 and 17 h (**E**,**F**), mouse grimace scale (**G**), and fecal corticosterone metabolites (**H**) were evaluated before (pre, overall range shaded gray), on the day of intervention (0), and the following recovery days (1–3). Statistical analysis was carried out with repeated measures two-way ANOVA, followed by Dunnett’s test for comparison to baseline within the models and Tukey’s test for inter-model differences. (* *p* ≤ 0.05, significant differences to the pre-values within each model for (a) orthotopic (n = 5–9), (b) subcutaneous (n = 7–9), (c) intravenous (n = 6–8)).

**Figure 3 biomedicines-12-01494-f003:**
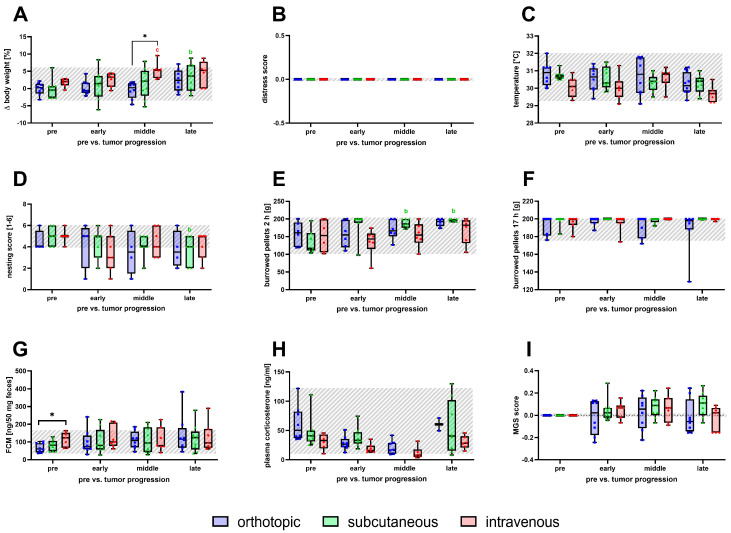
Animal welfare assessment during tumor progression for the orthotopic, subcutaneous, and intravenous pancreatic cancer model. Body weight change (**A**), distress score (**B**), perianal temperature (**C**), nesting behavior (**D**), burrowing activity after 2 and 17 h (**E**,**F**), concentration of fecal corticosterone metabolites (**G**), plasma corticosterone concentration (**H**), and mouse grimace scale (**I**) were evaluated before tumor cell injection (pre, overall range shaded gray) and during early, middle, and late phase of tumor progression. Statistical analysis was carried out with repeated measures two-way ANOVA (**A**,**C**–**F**) or mixed-effects model (**B**,**G**–**I**), followed by Dunnett’s test for comparison to baseline within the models and Tukey’s test for significance of inter-model differences. (* *p* ≤ 0.05, significant differences to the pre-values within each model for orthotopic (n = 4–9); (b) subcutaneous (n = 7–9), (c) intravenous (n = 6–8)).

**Figure 4 biomedicines-12-01494-f004:**
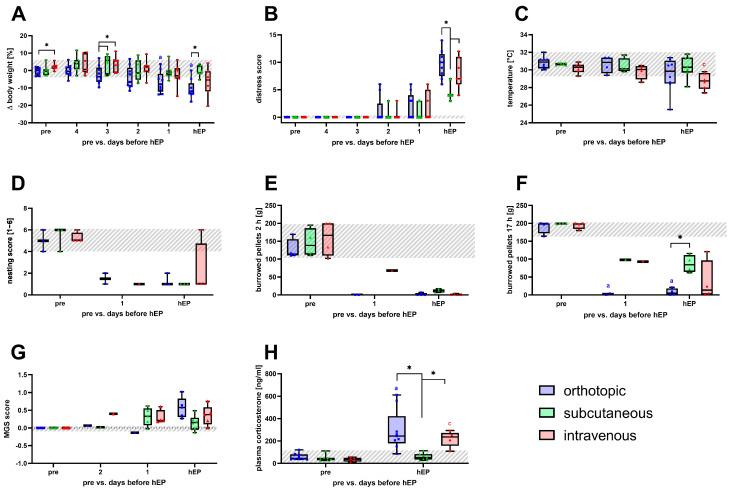
Severity assessment parameters are monitored until the individual humane endpoint in the orthotopic, subcutaneous, and intravenous pancreatic cancer model. Body weight change (**A**) and distress score (**B**) were continuously monitored before (overall range shaded gray) as well as on the four days before and the day of the humane endpoint (4, 3, 2, 1, and hEP). Perianal temperature (**C**), nesting (**D**), burrowing activity after 2 and 17 h (**E**,**F**) were assessed for the last two days (1, hEP). Mouse grimace scale (**G**) was evaluated from day two before the humane endpoint until the end of the experiment (2, 1, hEP). Plasma corticosterone concentration (**H**) was measured on the day of the individual humane endpoint. Statistical analysis was carried out with repeated measures two-way ANOVA (**A**,**B**,**H**) or mixed-effects model (**C**–**G**), followed by Dunnett’s test for comparison to baseline within the models and Tukey’s test for inter-model differences. (* *p* ≤ 0.05, significant differences to the pre-values within each model for (a) orthotopic (n = 3–9); subcutaneous (n = 3–7), (c) intravenous (n = 4–7)).

**Figure 5 biomedicines-12-01494-f005:**
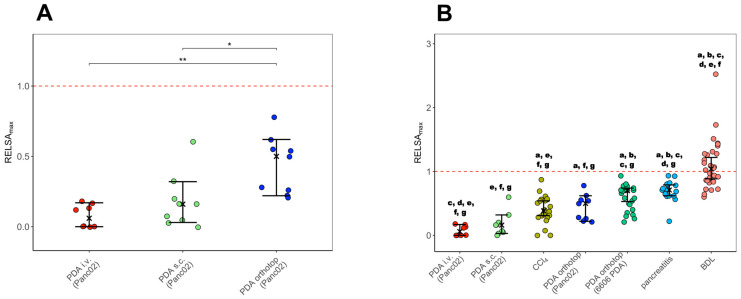
Between-model comparison of severity by RELSAmax. The distress of the intravenous (i.v.), subcutaneous (s.c.), and orthotopic (orthotop) pancreatic cancer models using the Panc02 cells was evaluated with the RELSAmax values. Significant differences between the RELSAmax estimates of the different models were evaluated by the Kruskal–Wallis test (Χ^2^ = 14.048, df = 2, *p*-value = 0.00089) (**A**). The RELSAmax estimates of the pancreatic cancer models were also compared to other murine gastrointestinal animal models such as CCl_4_ (model for liver fibrosis), PDA orthotop 6606PDA (orthotopic pancreatic cancer model with 6606PDA cells), pancreatitis, and BDL (bile duct ligation model). Significant differences between the RELSAmax estimates of the different models were evaluated by the Kruskal–Wallis test (Χ^2^ = 83.345, df = 6, *p*-value = <0.0001) (**B**). PDA i.v. (Panc02): n = 8 Ȓ = 0.075, CI95% = [−0.049;0.199]; PDA s.c. (Panc02): n = 9, Ȓ = 0.177, CI95% = [0.060;0.294]; PDA orthotop (Panc02):Ȓ = 0.440, CI95% = [0.323;0.557]; CCl4: n= 22, Ȓ = 0.401, CI95% = [0.293–0.510]; PDA orthotop (6606PDA): n = 26, Ȓ = 0.698, CI95% = [0.507;0.706]; pancreatitis: n= 16; Ȓ = 0.698, CI95% = [0.571;0.825]; BDL: n = 30, Ȓ = 1.102, CI95% = [1.009–1.194]. The graph displays RELSAmax values obtained from individual RELSA analyses, the bootstrapped median estimator, and the 95% confidence intervals. Significant differences indicated by (**A**): * *p* ≤ 0.05 and ** *p* ≤ 0.01; (**B**): significant different (*p* ≤ 0.05) to a (PDA i.v.), b (PDA s.c.), c (CCl4), d (PDA orthotop (Panc02), e (PDA orthotop 6606PDA), f (pancreatitis), and g (BDL).

## Data Availability

The raw data of the present study are attached in the form of an xlsx. file as Appendix A.

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
