# Peer review of "Evidence-Based Severity Assessment of Animal Models for Pancreatic Cancer"

_biomedicines, 2024, doi:10.3390/biomedicines12071494_

Round 1

Reviewer 1 Report

Comments and Suggestions for Authors

biomedicines-3044771

This paper by Schreiber et al. reports severity assessment of pancreatic cancer animals models using 3 different methods (orthotopic, subcutaneous, and intravenous injection of pancreatic cancer cells, Panc02 cells) during the progression stage. The authors found that the most important parameters were combined into a score and mapped against a reference data set by the Relative Severity Assessment procedure (RELSA) to obtain the maximum achieved severity for each animal (RELSAmax). They also revealed that a significantly higher RELSAmax for the orthotopic model than the subcutaneous and intravenous models. In addition, the pancreatic cancer models were less severe, when compared to other animal models, such as pancreatitis and bile duct ligation. I think this is a unique and interesting paper. However, there are too many parameters in experiments, although the number of animals used in the study was small. Specific comments are as follows:

1)     Abstract: Please define “3R” in line 18.

2)     The number of mice used was limited. Why?

3)     Please organize the text according to the aim of the study: (i) Comparison among 3 different cancer models, and (ii) Comparison among the cancer model, the non-cancer models (pancreatitis and bile duct ligation).

4)     The manuscript is relatively complex, please describe/state concisely in order to easily follow.

Reviewer 2 Report

Comments and Suggestions for Authors

Dear authors,

Knowing that a PubMed search for "animal models for pancreatic cancer" yields 5 published review articles in 2022-2024, it is important to understand the innovative perspective and added value that this review provides, compared to its predecessors. For example:

1 - Salu P, Reindl KM. Advancements in Preclinical Models of Pancreatic Cancer. Pancreas. 2024;53(2):e205-e220. doi:10.1097/MPA.0000000000002277.

2 - Forsythe SD, Pu T, Andrews SG, Madigan JP, Sadowski SM. Models in Pancreatic Neuroendocrine Neoplasms: Current Perspectives and Future Directions. Cancers (Basel). 2023;15(15):3756. Published 2023 Jul 25. doi:10.3390/cancers15153756.

3 - Vudatha V, Herremans KM, Freudenberger DC, Liu C, Trevino JG. In vivo models of pancreatic ductal adenocarcinoma. Adv Cancer Res. 2023;159:75-112. doi:10.1016/bs.acr.2023.02.002.

4 - Greco L, Rubbino F, Laghi L. Epithelial to Mesenchymal Transition as Mechanism of Progression of Pancreatic Cancer: From Mice to Men. Cancers (Basel). 2022;14(23):5797. Published 2022 Nov 24. doi:10.3390/cancers14235797.

5 - Gaspar TB, Lopes JM, Soares P, Vinagre J. An update on genetically engineered mouse models of pancreatic neuroendocrine neoplasms. Endocr Relat Cancer. 2022;29(12):R191-R208. Published 2022 Nov 2. doi:10.1530/ERC-22-0166.

In this scenario, the aim of the present study was to do a data-based severity assessment in different cell-derived murine cancer models, emphasizing the importance of refinement in research involving particularly challenging models.

However, after carefully reading the paper you submitted, I have encountered some doubts, so to better understand your work, I would appreciate it if you could clarify the following questions:

1) Introduction: it would be helpful for the reader, in addition to the objective, to explain the benefits that this type of severity assessment approach can bring to animal welfare and researchers.

2) Materials and Methods:

* This work includes the description of many different protocols, which are useful for comparison with the main data related to pancreatic cancer models. Despite there being many protocols with various details, I find the reading not difficult. I would just like the number of animals included in each protocol to be more obvious.

* For each protocol, please include the number of animals found deceased (if any) and those that were euthanized following the application of humane endpoints versus those that were euthanized because the protocol had ended (but did not meet euthanasia criteria according to the humane endpoints).

2.3 Animal models for pancreatic cancer

- How long were mice kept single-housed?

- Were the sutures performed continuous or interrupted? When you mention that "the subcutaneous skin was sewed with a 5-0 prolene suture" are you referring to a final intradermal suture?

2.8 Assessment of behavioral parameters

- What was the rational for excluding from the final data analysis animals that burrowed less than 100 g pellets within 2 hours and less than 150g overnight during the pre-phase? Please include references.

3) Results:

* The figures contain small-sized graphs (understandable, to make the overall view readable) that lose quality when enlarged. Please review the image quality.

4) Discussion:

4.1 Applicability of the RELSAmax for severity assessment

- Quantifying the highest experienced distress of mice in an animal model is very important, but it can represent a specific moment in the protocol that the animal is able to overcome (with analgesia, for example). It may not reflect the overall discomfort the animal experiences and it does not predict the earlier or later achievement of HEP. Please comment.

4.2 Severity assessment of cell-induced PDA models

- Did you notice any changes in fecal production in the PDA orthotopic model? Laparotomy can induce a reduction in GI motility, decreasing animal welfare.

- Considering that the parameters most influencing the decision to terminate the protocol due to reaching the HEP differ for the three PDA models, shouldn’t it also be emphasized that in the case of the orthotopic model, animals reach the HEP earlier (~50 days versus ~80 days with the other models, figure S1), indicating greater aggressiveness of this model and, therefore, requiring closer supervision?

5) Conclusion:

- Consider the possibility of transferring this sentence from the Discussion to the Conclusion section “Despite the varying severity of differently induced cancer models, the correct choice of an animal model should rely on the specific research question and the optimal transfer from bench to bedside.” Otherwise, why should we use animals in research?

Thank you for the work you've presented.

I wish you the best of luck and continued success.

Round 2

Reviewer 1 Report

Comments and Suggestions for Authors

The revised manuscript has been greatly improved and the response to the comments is adequate. It is now acceptable in the journal.